# Cadmium-induced DAG-PKC/IP$_3$-Ca$^{2+}$ double signaling pathway and cell injury in gill of freshwater crabs *Sinopotamon henanense*

**Jianxia Liu**[1,2], **Lijuan Yu**[3], **Jiachuan Yang**[1,2], **Rui Zhang**[1,2], **Jing Bai**[1,2], **Jinxiang Wang**[1,2]*

**1** College of Agronomy and Life Science, Shanxi Datong University, Datong, Shanxi, China, **2** Research and Development Center of Agricultural Facility Technology, Shanxi Datong University, Datong, China, **3** Agro-Products Processing Research Institute, Yunnan Academy of Agricultural Sciences, Kunming, Yunnan, China

* wangjx11@163.com

## Abstract

Cadmium (Cd) is an extremely toxic heavy metal and widespread in the environment, which has been demonstrated closely associated with numerous organs damage and even death. To evaluate the signaling pathway and the toxicity induced by Cd$^{2+}$ in aquatic organisms, freshwater crabs *Sinopotamon henanense* were exposed to Cd$^{2+}$ (0, 14.5, 29 and 58 mg/L) for 4 d. The results showed that Cd$^{2+}$ induced the increase of diacylglycerol (DAG) and inositol-1,4,5-triphosphate (IP$_3$), the rapid change of Ca$^{2+}$ concentration in the cytoplasm and activation of protein kinase C (PKC) in the gill of crab. Also, Cd$^{2+}$ raised the expressions of 6 genes relating with protein detoxification and folding, including cytochrome P450 (CYP450), BAG regulator 3 (BAG3), protein disulfide isomerase (PDI) and 3 genes encoding heat shock proteins (Hsp21, Hsp27 and Hsp70). And, it was observed that Cd$^{2+}$ exposure aggravated filament swelling and raised the number of dead epidermal cell in gill. Meanwhile, Cd$^{2+}$ exposure degraded DNA which was accompanied by an obvious decrease in ATP level. In conclusion, Cd$^{2+}$ triggered DAG-PKC/IP$_3$-Ca$^{2+}$ double signaling pathway, caused gene expression, DNA degradation and ATP decrease, and finally induced the cell injury in *S. henanense* based on histological observation. This study would provide us fundamental information to discover the underlying mechanism of Cd$^{2+}$-induced cytotoxicity.

## 1. Introduction

Cadmium ion (Cd$^{2+}$) is an extremely toxic industrial pollution of global concern. Due to the thriving coal mining, coking plant, iron, steel, and chemical industries, highly toxic Cd$^{2+}$ have been frequently detected in rivers, estuaries, and near-shore waters. During the heavy metal pollution incident in Longjiang, Guangxi, China in 2012, Cd$^{2+}$ concentration in the water reached a peak that exceeded the standard by 80 times, and the length of the river section where Cd$^{2+}$ concentration exceeded the standard

**Data availability statement:** All relevant data are within the manuscript and its Supporting Information files.

**Funding:** The work was supported by the Natural Science Foundation of Shanxi Province (201701D121084 to JW), the Applied Basic Research Project of Datong City (2022053 to JL), and the Shanxi Province Science Foundation for Youths (20240302121202 to JY).

**Competing interests:** The authors declare that they have no known competing financial interests or personal relationships that could have appeared to influence the work reported in this paper.

by more than 5 times was 100 kilometers [1]. $Cd^{2+}$ couldn't be decomposed through chemical or microbial pathway and would continuously and irreversibly accumulate in aquatic animals through feeding and respiration, posing a great threat to public health worldwide [2]. $Cd^{2+}$ disturbs both cell membrane integrity and various metabolic responses, and exerts toxic effects on multiple organs [3–6]. The dietary and water polluted with $Cd^{2+}$ can not only lead to distinct damages to the outer skin, scales, and gills of goldfish, but also causes inner damages in different organs such as gills lamella shrunk and broken, the necrotic muscles myoepithelium along with disrupted muscles bundle, and abnormal dilations in the liver cells [7]. $Cd^{2+}$ also caused the decrease of respiratory function in the freshwater crab *Barytelphusa gureini*, gill hyperplasia and degenerative changes in visceral organs, elevation in MDA and attenuation in GSH activity in *Oreochromis niloticus* [8,9]. Exposure to $Cd^{2+}$ led to the changes of protein metabolism and energy imbalances with a decrease in tissue glycogen and an elevated haemolymph glucose in snail [10].

A series of toxic symptoms caused by $Cd^{2+}$ accumulation are associated with oxidant damage, protein misfold, induction and inhibition of enzymes, which are known to trigger some unique signaling cascades from the membrane to the nucleus, being characterized by enhanced gene expression of antioxidant enzymes, detoxification proteins and molecular chaperone, to adapt themselves to $Cd^{2+}$-contaminated environments [11,12]. Increasing evidences showed that exposure to $Cd^{2+}$ could induce various cellular responses such as autophagy, apoptosis or necrosis. Autophagic cell death was restrained, but apoptosis was mildly induced in NRK52E cells exposed to $Cd^{2+}$, which process is regulated by activation of protein kinase C (PKC) [13]. In the renal proximal tubule cells, $Cd^{2+}$ induced apoptotic cell death and intracellular calcium ion ($Ca^{2+}$) overload is involved in this process [14]. $Cd^{2+}$ induced autophagy and apoptosis through elevation of cytosolic $Ca^{2+}$ concentration ($[Ca^{2+}]c$) in MES-13 cells [4]. The above studies have also strongly shown an association between $Cd^{2+}$ stress and the increase in $[Ca^{2+}]c$. The $[Ca^{2+}]c$ is tightly controlled by intracellular $Ca^{2+}$ stores such as endoplasmic reticulum (ER), mitochondria and lysosome, which are also the main sites of autophagic and apoptotic regulation. Phospholipase C (PLC) is a lipid-anchored membrane protein and cleaves phosphatidylinositol-4,5-bisphosphate ($PIP_2$) to form inositol-1,4,5-triphosphate ($IP_3$) and diacylglycerol (DAG). $IP_3$ leads to the release of $Ca^{2+}$ from ER by interacting with $IP_3$ receptors ($IP_3Rs$) which is the main $Ca^{2+}$-release channels in the ER. DAG, along with $Ca^{2+}$ signal, induces PKC phosphorylation. At the same time, $IP_3Rs$ can transfer $Ca^{2+}$ to the mitochondria, thereby increasing apoptosis sensitivity and inhibiting basal autophagy. Gu et al. [16] reported that $Cd^{2+}$ induces $Ca^{2+}$ release from ER, associated with ER stress through PLC-$IP_3$ signaling pathway. $Ca^{2+}$-release from the ER ultimately leads to $[Ca^{2+}]c$ elevation and apoptotic death [14].

*Sinopotamon henanence*, a Crustacea, is widely distributed in the Qinhe rivers in China. As a regional dominant species, it is sensitive to heavy metals and is often used for toxicity assessment in aquatic environments. Previous studies indicated that waterborne $Cd^{2+}$ pollution causes $Cd^{2+}$ accumulation in target organs, including the gills, hepatopancreas, antennal gland, and ovary, with the highest level in the gills [17]. $Cd^{2+}$ induced production of reactive oxygen species, lipid peroxidation, metabolic

changes and various organs damage in *S. henanense* [17–19]. Additionally, $Cd^{2+}$ induced cell death in the hepatopancreas and gills [4,20]. $Cd^{2+}$ significantly increased the $[Ca^{2+}]c$, $Ca^{2+}$-ATPase activity, PKC activity and caspase-9/caspase-3 activities, and $Ca^{2+}$ played key roles in $Cd^{2+}$-induced apoptosis through a mitochondrial pathway [4,17,21]. However, the source of elevated $[Ca^{2+}]c$, adaptive responses, and the molecular mechanisms underlying $Cd^{2+}$-induced cell death are still unknown. The objective of this study was to examine the cellular signaling pathways and physiological responses as induced by $Cd^{2+}$ in the gill of freshwater crab. This work would provide an important experimental evidence for clarifying DAG-PKC/$IP_3$-$Ca^{2+}$ signal pathway activated by $Cd^{2+}$ roles in intracellular protective responses through regulating the expressions of CYP450, BAG3, PDI and HSPs in the early period of stress and cell apoptosis in the late stage of stress.

## 2. Materials and methods

### 2.1. Experimental animal and treatment

Freshwater crabs, *S.henanense*, were purchased from the Dongan Aquatic Market in Taiyuan city (Shanxi Province, China). First, healthy male crabs in similar size and weight (18.0±0.5 g) were acclimated for 2 weeks in glass aquaria filled with city tap water (pH 7.5, dechlorinated with dissolved oxygen of 8.0–8.3 mg/L) at the temperature of 20±2℃ and were fed three times a week.

After 2 weeks, the crabs were randomly divided into four equal groups and treated with $Cd^{2+}$ solution at three sublethal concentrations (14.5, 29 and 58 mg/L) for 96 h, corresponding to 6.25%, 12.5% and 25% of the 96-h $LC_{50}$, based on the 96 h $LC_{50}$ of $Cd^{2+}$ for crabs (232 mg/L) [15], respectively with the tap water as control. In this study, the crabs were denied food, and all other culture conditions were kept as same as those used during the acclimation period.

### 2.2. Sample preparation

After $Cd^{2+}$ exposure, second messengers in cells are rapidly generated following activation of membrane receptors. Next, some genes are transcribed and translated through the activation of a series of signaling proteins, such as PKC phosphorylation. Ultimately, the biological effects of the cell, including changes in tissue structure and DNA degradation, are manifested. Therefore, in this experiment, to detect DAG and $IP_3$ contents, the gill tissues were immediately excised from the crabs exposed to $Cd^{2+}$ for 1, 2, 4 and 8 h. To analyze PKC activity and extract RNA, the gill tissues were excised after the crabs were exposed to $Cd^{2+}$ for 12, 24 and 48 h. A piece of gill from each individual crab was fixed for histological observation after $Cd^{2+}$ exposure for 48, 72 and 96 h, and the reminds were used to extract DNA and ATP content measurement. Because the effects of continuous $Cd^{2+}$ exposure on the histological structure of gills and cell death were found to be concentration-dependent, crabs exposed to 58 mg/L $Cd^{2+}$ was used in the present study to analyze the cellular signaling pathways activated by $Cd^{2+}$, including PKC activation, gene expression, changes in tissue structure and energy, and DNA injury [19,20].

### 2.3. Measurement of DAG and $IP_3$ contents

DAG and $IP_3$ contents were determined using Elisa Kits from the MLbio Biotech Co. (Shanghai, China) according to corresponding instructions. Tissue pieces were rinsed with ice-cold PBS (0.01M, pH7.4), weighed and then minced into small pieces which were homogenized in PBS (1:9 g:ml) on ice. The homogenates were centrifugated for 10 min at 5000×g and the supernatants were collected and measured absorbance at 450 nm. The DAG/$IP_3$ amount in each sample was calculated according to the standard curve (y=2771.8x-118.49 for $IP_3$, y=9.9582x-0.5996 for DAG, respectively).

### 2.4. Isolation of gill cell and detection of $Ca^{2+}$ concentration

Single cell was isolated from the untreated crab gill using trypsin digestion, and then loaded with 5 μM Fluo-3/AM (Beyotime Institute of Biotechnology, Shanghai, China) for 30 min at 37 ℃ in the dark. After incubation, the suspension cells were washed twice with PBS. Then the cells were treated with 5.8 mg/L $Cd^{2+}$. Fluorescent probe was excited at 488 nm,

 

and emission fluorescent was filtered at 525 nm. The images were recorded in 60 min with a fluorescence microscopea (Olympus, Japan) and laser confocal microscope (LSM 510 META, Leica, Bensheim, Germany).

## 2.5. Measurement of PKC activity

PKC activity was measured with PepTag assay kit (Promega Co. USA). The reactive mixture with a final volume of 25 mL consisted of 5 mL reaction buffer, 5 mL PepTag C1 (0.4 mg/mL), 5 mL PKC activator solution, 1 mL peptide protection solution and 9 mL sample. After incubation for 30 min, the reaction was stopped by heating under 95 °C for 10 min. SDS PAGE electrophoresis of 0.8% agarose gel at 100 V for 15 min was performed to separate the mixture. After electrophoresis, the bands of phosphorylated PepTag C1 peptide were scanned and the band densities were quantified using the AlphaView analysis software (Fluorchem HD2 Chemilumilescent, Fluorescent and Visible Light Gel Imaging System, Alpha Innotech, USA).

## 2.6. Real-time quantitative PCR

Total RNA was isolated from gills using TRIzol Reagent (BBI CO., LTD., China). Subsequently, about 1 μg of total RNA were reverse transcribed using random primer $p(dN)_6$ and Maxin reverse transcriptase. And, real-time quantification PCR analysis was performed using the SYBRGreen kit (BBI CO., LTD., China) according to the manufacturer's instructions with following primers (Table 1). The RT-qPCR reaction mixture with a final volume of 20 μL consisted of 10 μL of 2×SYBR Green Master Mix, 0.2 μM of each forward and reverse primers, 2 μL of 10×diluted cDNA template, and water. PCR reactions were performed under the conditions: heating at 95 °C for 3 min, followed by 45 cycles of heating at 95 °C for 5 s and annealing at 60 °C for 30 s. Relative expressions were calculated using the $2^{-(\Delta\Delta Ct)}$ method with *Rpl38* as the reference gene. Three biological replications with four technical repetitions were performed on each sample.

## 2.7. Histological observation

Gill tissues were fixed for 24 h at room temperature in 4% paraformaldehyde buffer, followed by dehydration with ethanol and toluene in a series gradient and embedded in paraffin. Serial sections with the thick of about 4 μm were obtained and stained with hematoxylin and eosin before observation with a light microscope (Olympus BX51, Tokyo, Japan).

## 2.8. DNA-fragmentation assay

DNA injury was analyzed by assessing DNA fragmentation using the DNA purification kit (Beyotime Institute of Biotechnology, Shanghai, China). Firstly, 20 mg of the sample was collected. Then the sample was minced and lysed in a lysis buffer containing proteinase K, followed by centrifugation. The mixture was then eluted twice with an elution buffer, and subsequently, DNA was eluted using deionized water or a hydrophilic elution buffer. The eluates containing DNA were separated with the electrophoresis system of 1.5% agarose gel at 100 V for 1 h. DNA bands were visualized and captured by a ultraviolet gel documentation system.

**Table 1. Primer sequences used for real-time PCR analysis.**

| Gene | Forward primer | Reverse primer |
| --- | --- | --- |
| *CYP450* | 5′-GAACTTCTCAAGGCATTCATCG-3′ | 5′-ATTCGGCGGTGTCTTTGAGT-3′ |
| *BAG3* | 5′-CAGATCGAAAGGGTGAGTGG-3′ | 5′-TTGCTGATGGCATTGTTGAT-3′ |
| *PDI* | 5′-GAGGGCAAGAGCAAGTATGAA-3′ | 5′-ATCATAGTGACCAGCCTCCTTT-3′ |
| *HSP21* | 5′-AGAACCAAGCAGTGTCCCTCA-3′ | 5′-CCCACTCTTCTGTTCCACGAT-3′ |
| *HSP27* | 5′-CGAGCAGAAGGAAGGAAACA −3′ | 5′-AGCGTTGATGGTCAGCACA −3′ |
| *HSP70* | 5′-ATTTCTACACCTCCGTCACCC-3′ | 5′-CTTTGTCCATCTTGGCATCAC-3′ |
| *RpL38* | 5′-ATGACATTCTTCTGCTTATTGGC-3′ | 5′-ATGCTTCCCACCTCTTCGTT-3′ |

## 2.9. Determination of adenosine triphosphate (ATP) content

Total ATP content was measured based on creatine phosphate generated from ATP and creatine catalyzed by creatine kinase with a ATP assay kit (Nanjing Jiancheng Institute of Bio-Engineering, China). Protein content was determined according to the method reported by Bradford [22] with bovine serum albumin as the standard.

## 2.10. Statistical analysis

Statistical analyses were performed with SPSS 17.0 software. And, results were expressed as means ± SE (n = 3) of three biological replicates per treatment group. Statistical analysis was carried out using one-way analysis of variance (ANOVA) to evaluate whether the means were significantly different, taking $P < 0.05$ as minimally significant.

## 3. Results

### 3.1. $Cd^{2+}$ increased DAG and $IP_3$ contents in *S. henanense* gill

The contents of DAG and $IP_3$ in the gill were determined after the freshwater crab, *S. henanense* were exposed to $Cd^{2+}$ (Fig 1). As shown in Fig 1A, 58 mg/L $Cd^{2+}$ induced that DAG content increased within 1 h, reached the maximum of 36.5 ng/g after 4 h, and slightly decreased to 28.5 ng/g after 8 h. And as shown in Fig. 1B, the effect of $Cd^{2+}$ on DAG content was significantly related with its concentration, when $Cd^{2+}$ increased from 14.5 to 58 mg/L, gill DAG contents increased from 14.8 to 34.1 ng/g. Meanwhile, as it can be seen from Fig 1C and Fig 1D, the $IP_3$ content changed with the same trend as that of DAG. $IP_3$ content significantly increased with the exposure time at the early and peaked at 7.2 ng/g after being exposed for 4 h (Fig 1C), and there was an obvious positive relationship between $IP_3$ content and $Cd^{2+}$ concentration (Fig 1D). These results supported that $Cd^{2+}$ exposure induced the increase of DAG and $IP_3$ in *S. henanense* gill, and the inducing effect was dependent on both $Cd^{2+}$ exposure time and concentration.

### 3.2. $Cd^{2+}$ rapidly increased $[Ca^{2+}]c$ in *S. henanense* gill

Rapid change of $[Ca^{2+}]c$ is an important signal regulating multi biological processes, thus the related change was observed in *S. henanense* gill exposed to $Cd^{2+}$ and shown in Fig 2. As shown in Fig 2A, there was almost no fluorescence in the gill cell at the initial exposure, indicated that $[Ca^{2+}]c$ was very low and almost close to zero (Fig 2A). However, combing Fig 2A to Fig 2C, the $[Ca^{2+}]c$ gradually increased during exposure for 0–15 min (Fig 2A-B) but soared at 30 min (Fig 2C). And, according to Fig 2D and Fig 2E, $[Ca^{2+}]c$ obviously reduced (Fig 2D) after 45 min and almost disappeared at 60 min (Fig 2E). Totally, $Cd^{2+}$ induced the $[Ca^{2+}]c$ rapidly increased in gill cell which only lasted for 30 min. Moreover, with a laser confocal microscope, it was observed that the fluorescence in the cytoplasm extended from the endoplasmic reticulum at cell core to the whole cell after gill cell being exposed to $Cd^{2+}$ for 15 min (Fig 2F-G). This result further confirmed that $Cd^{2+}$ induced the rapid increase of $[Ca^{2+}]c$, and indicated that endoplasmic reticulum might be the source of $Ca^{2+}$.

### 3.3. $Cd^{2+}$ increased PKC activity in *S. henanense* gill

PKC, a class of serine/threonine kinases, amplify signal and modulate gene expression in differentiation and proliferation events by phosphorylating a series of transcription factors, such as API transcription factor, NF-κBa transcription factor [23–25]. Thus, PKC activity in *S. henanense* gill exposed to 58 mg/L $Cd^{2+}$ was analyzed and PKC activity in each group is shown as the increasing multiple compared to control, which has been set as an value of 1 (Fig 3). $Cd^{2+}$ induced that PKC activity significantly increased, reached to the highest value at 24 h which was about 3.8-fold of control, and then decreased gradually at 48 h. Finally, PKC activity was still signifcanttly higher than that of control ($P < 0.01$) although there was a gradual decrease.

### 3.4. $Cd^{2+}$ increased the expression of 6 genes relating with protein-folding

$Cd^{2+}$ exposure induced the increase of PKC activity in *S. henanense* gill and enlightened us to consider if the increase of PKC activity trigger downstream signal pathway, such as the expression of the genes relating with

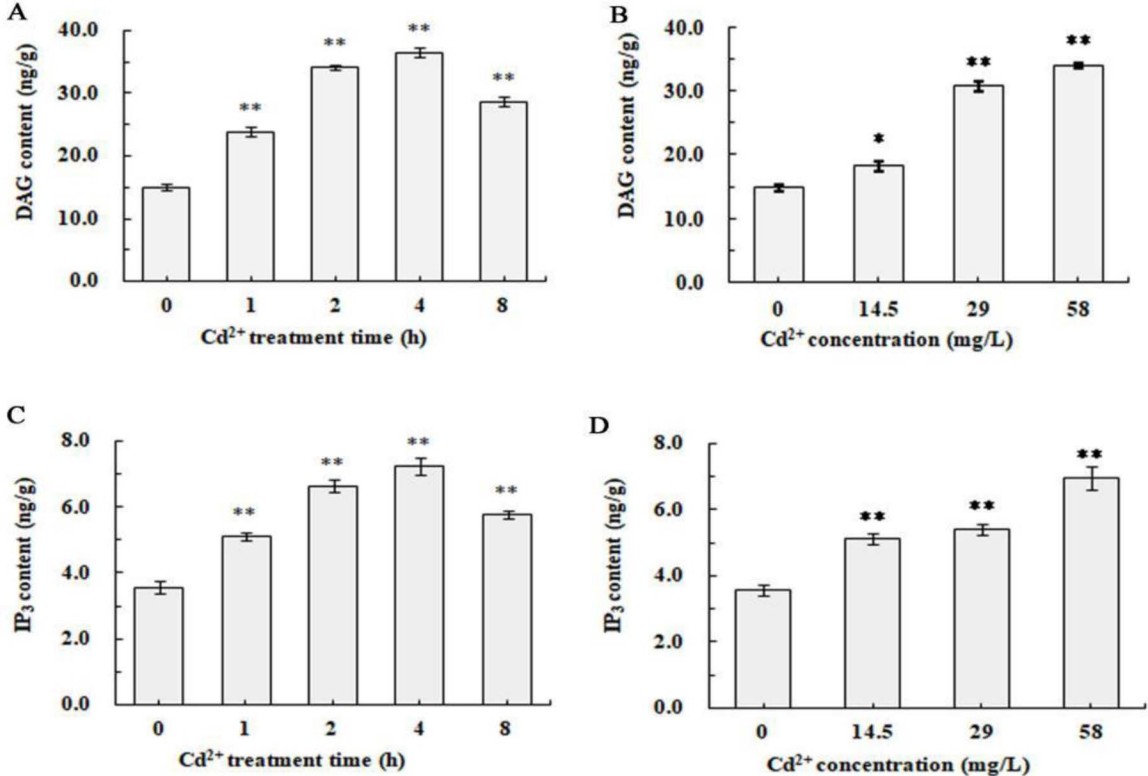

**Fig 1. Effects of acute Cd²⁺ exposure on DAG and IP₃ contents in *S. henanense* gill.** The crabs were treated with 58 mg/L Cd²⁺ **(A, C)**; or 14.5, 29 and 58 mg/L Cd²⁺ for 2 h **(B, D)**. Asterisks indicate a significant difference to the control (*$P < 0.05$, **$P < 0.01$).

protein-folding. So the expression of 6 genes in *S. henanense* gill exposed to 58 mg/L Cd²⁺ were quantified by qRT-PCR method, which relate with protein-folding including Cytochrome P450 (CYP450), BAG regulator 3 (BAG3), protein disulfide isomerase (PDI), heat shock protein 21 (Hsp21), heat shock protein 27 (Hsp27) and heat shock protein 70 (Hsp70) (Fig 4). The expression of 6 genes changed in a similar trend, which continuously climbed when the *S. henanense* was exposed to Cd²⁺ for 0–24 h and then gradually decreased. Exposure to Cd²⁺ for 24 h raised the expression of CYP450, BAG3, PDI, Hsp21, Hsp27 and Hsp70 by 4.9, 3.7, 3.1, 2.3, 2.9, 2.3 times (*$P < 0.01$), respectively. And, at the final of the exposure, the expression of 6 genes in *S. henanense* gill were still obviously higher than that of control, although which gradually decreased form the peak value. Notably, we found that the change of the expression of the 6 genes was consistent with that of PKC activity when combing Fig 3 and Fig 4. This indicated that Cd²⁺ exposure induced the increase of PKC activity in *S. henanense* gill, which might trigger the expression of genes relating with protein-folding in turn.

### 3.5. Cd²⁺ destructed the structure of *S.henanense* gill

The gill lamellae of *S. henanense* consisted of cuticle, epithelial cells and gill cavity. Here, these three tissues in *S.henanense* exposed to 58 mg/L Cd²⁺ were observed (Fig 5A-D). As Fig 5A displayed, the flat slender gill filaments, narrow gill cavity and more epidermal cells at the junction between the upper and lower wall in gill lamellae were observed in the control group, and the nuclei of epidermal cells was round or oval dyed with hematoxylin. However, Cd²⁺ exposure destroyed gill filament structure and cause the cell death (Fig 5B-D). After exposure for 48 h, gill filament was irregularly thickened,

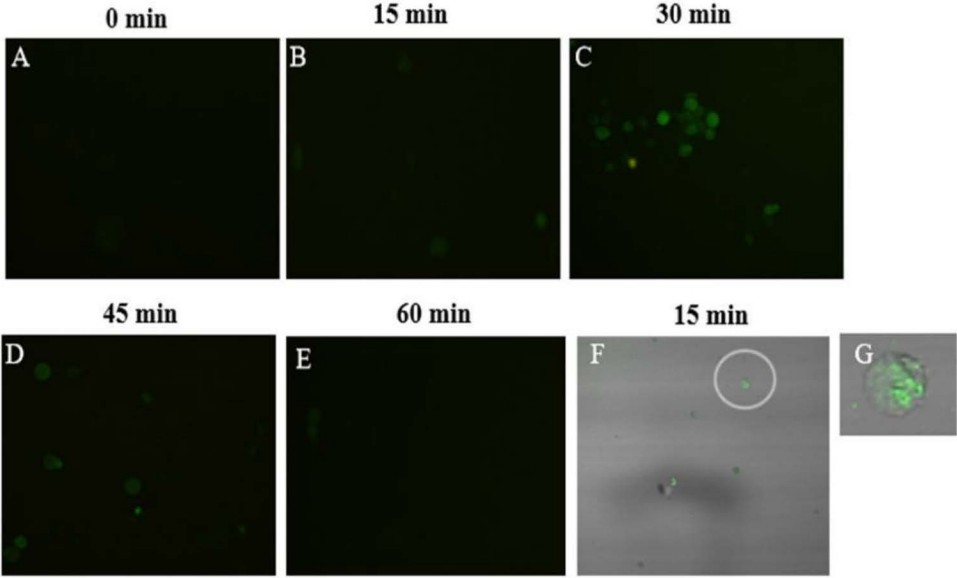

**Fig 2. Effects of acute Cd²⁺ exposure on [Ca²⁺]c in gill cells of *S. henanense*.** A-E: the fluorescent images (400×); F and G: the laser confocal fluorescent images (F, 400×; G, the cell in Fig. 2F magnified by 6 times).

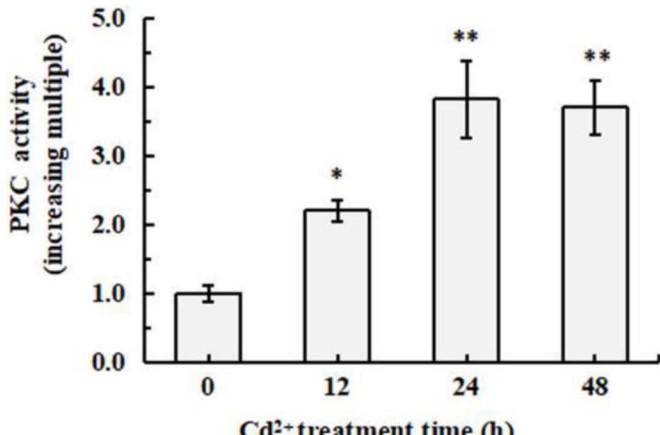

**Fig 3. Effects of Cd²⁺ exposure on PKC activity in *S. henanense* gill.** Asterisks indicate a significant difference to the control (*$P < 0.05$, **$P < 0.01$).

gill cavity was enlarged, and some epidermal cells at the junction disappeared while some small and irregular residual body appeared (labeled with red triangle in Fig 5B). After 72 and 96 h, the gill filament was further damaged culminating in their breakdown (yellow asterisk in Fig 5C-D), and more epidermal cells disappeared (red arrow in Fig 5C-D).

### 3.6. Cd²⁺ degraded DNA and decreased ATP content

The DNA integrity and ATP level (Fig 6A-B)showed that DNA was intact in the control (0 h), but obviously degraded to ladder after being exposed to Cd²⁺ for 48 h and completely dispersed after 96 h. As Fig 6B displayed, Cd²⁺ exposure decreased ATP level from 387 to 189 µmol/g.

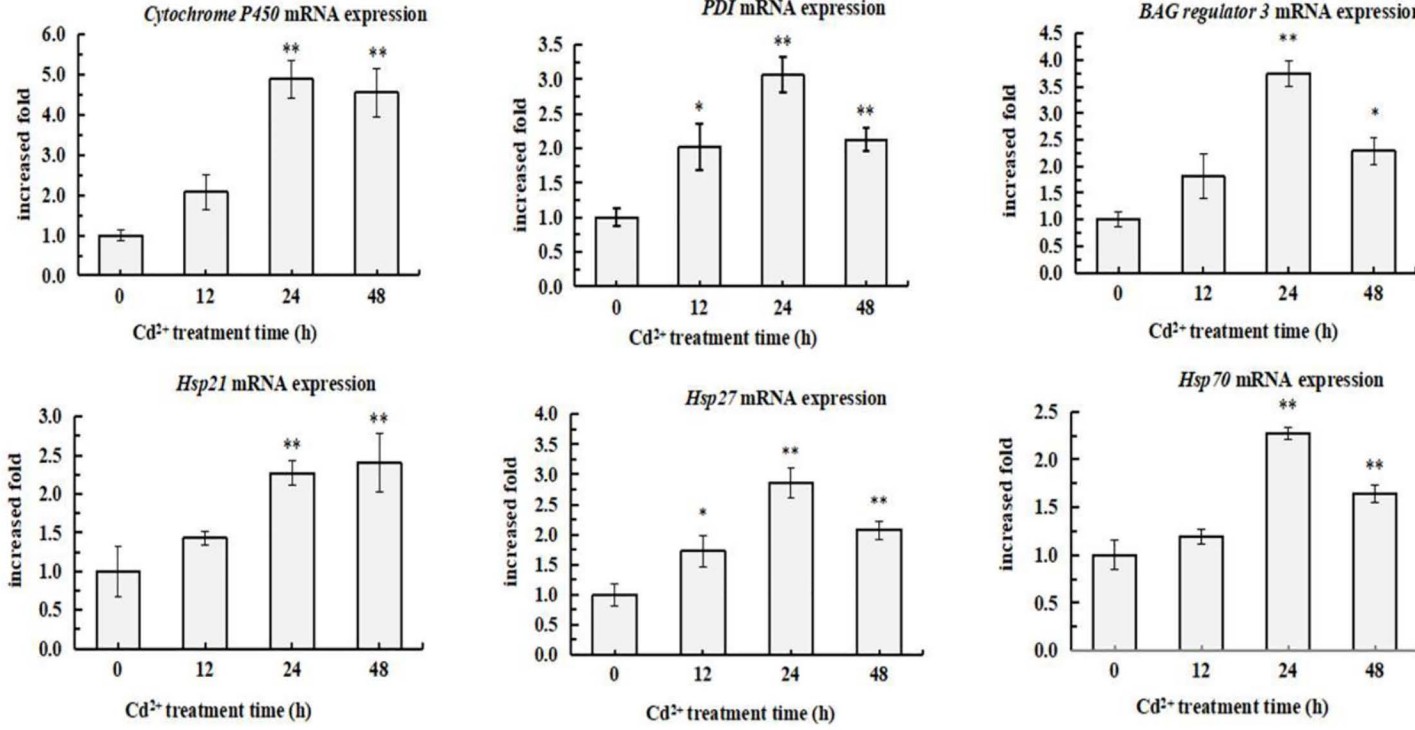

**Fig 4. Effects of Cd²⁺ exposure on genes expression in the gill of *S. henanense.*** Crabs were treated with 58 mg/L Cd²⁺ for 0, 12, 24, 48h. Asterisks indicate a significant difference to the control (*$P < 0.05$, **$P < 0.01$).

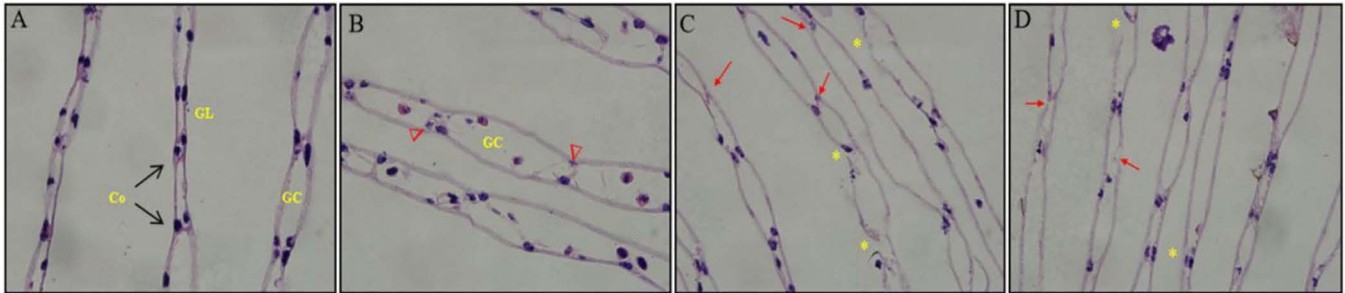

**Fig 5. Effects of Cd²⁺ exposure on gill of *S. henanense.*** A-D: HE-stained gill section, 600×; A: control; B: exposure to Cd²⁺ for 48 h; C: exposure to Cd²⁺ for 72 **h.** D: exposure to Cd²⁺ for 96 **h.** (Co: connection of gill lamellae; GC: gill cavity; GL: gill lamellae; red triangle: small and irregular residual body; yellow asterisk: broken gill filaments; red arrow: disappeared cells).

## 4. Discussion

### 4.1. DAG-PKC and IP₃-Ca²⁺ signaling pathway

Cd²⁺ is a highly toxic metal that cause oxidative damage, metabolic disorders, and dysfunction in aquatic organisms. However, complex special signaling might be activated in Cd²⁺-resisting species to resist Cd²⁺ and reduce its toxicity. DAG and IP₃ are two important intracellular signaling molecules, which are catalyzed by PLC. IP₃Rs located on the ER are activated in response to IP₃, which leads to the release of Ca²⁺ from ER and triggers Ca²⁺ signal pathway. Ca²⁺ is a

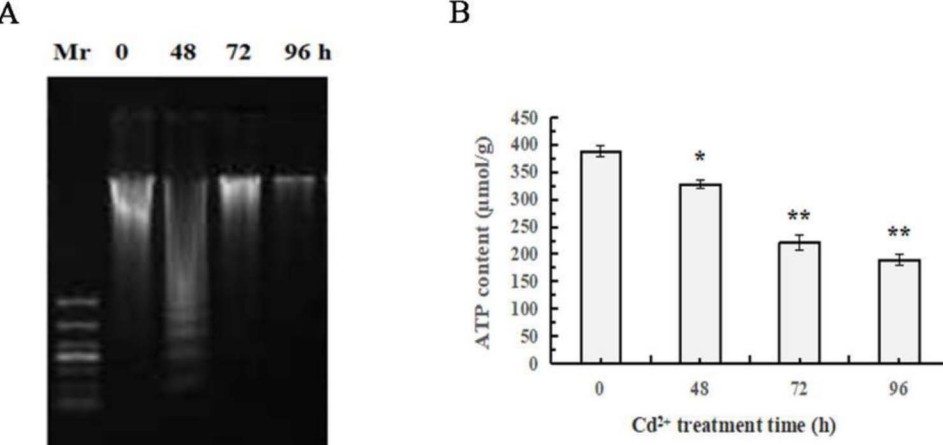

**Fig 6. Effects of Cd²⁺ exposure on DNA integrity and ATP level in *S. henanense* gill.** A: DNA fragmentation of gill cells (Mr: DNA marker); B: ATP level (n = 3, *$P < 0.05$, **$P < 0.01$).

versatile and dynamic second messenger that regulates biological processes through rapid change of [Ca²⁺]c, including secretion, motility, apoptosis and autophagy [26,27]. The generation of DAG and IP₃ and the following elevated [Ca²⁺] c promote the binding of Ca²⁺ to PKC and translocation from cytoplasm to cell membrane, where PKC is activated by DAG [28]. Actived PKC phosphorylate various transcription factors, such as API and NF-κB, modulating gene expression during differentiation and proliferation events, also phosphorylates other signal-transducing kinases, such as S6 kinase and Raf kinase, participating in translation process and mitogenesis [23–25]. In resting conditions, many cells display a constitutively low level of DAG/IP₃-mediated signaling. Lawal et al. [26] reported that intracellular Ca²⁺ levels increased significantly in HEK 293 cells after being exposed to Cd²⁺ for 24h. Cd²⁺-elevated [Ca²⁺]c were mainly originated from IP₃R-mediated ER which released Ca²⁺, but not from extracellular medium. U73122, the PLC-specific inhibitor, prevented the Cd²⁺-dependent increase in Ca²⁺ level and abolished Cd²⁺-dependent caspase 3 activation as well as apoptosis [29]. Moreover, in Cd²⁺-treated rPT cells, elevations of [Ca²⁺]c and mitochondrial Ca²⁺ concentration with depletion of ER Ca²⁺ levels occurred at the same time. Cytosolic Ca²⁺ overload originated from IP₃R-mediated ER Ca²⁺ release has a negative impact on Cd²⁺ nephrotoxicity through its promotion of apoptosis [14]. Previous experiments supported that Cd²⁺ induced elevation of [Ca²⁺]c and participated in the process of gill cell apoptosis in *S.henanense* by regulating the activities of caspase 9 and caspase 3, but source of Ca²⁺ in stress and its upstream signal moleculars are still unclear [21]. In this study, Cd²⁺ increased the contents of both DAG and IP₃ in gill tissue at the early state of stress (1~8 h), and induced the rapid change of [Ca²⁺]c. Both DAG and IP₃ are transient metabolites produced by PIP₂ hydrolysis. After activating the signaling pathway, IP₃ will be converted into inactive IP₂ and IP₁ through dephosphorylated by phosphatases, and DAG will either be phosphorylated to form phosphatidic acid (PA) or hydrolyzed by lipases into glycerol and fatty acids, thus becoming inactivated. Therefore, the contents of DAG and IP₃ reach their peaks at 4 h after Cd²⁺ exposure and then gradually decrease.

Elevated IP₃ level and figure of Ca²⁺ sparking based on laser confocal microscope (Fig 2G) clearly confirmed that Cd²⁺-induced IP₃-mediated ER Ca²⁺ pool release Ca²⁺. At 15 min of Cd²⁺ exposure, Ca²⁺ are rapidly released from ER into cytoplasm, resulting in the increase of [Ca²⁺]c. After 45 min of stress, Ca²⁺ is transported out of the cytoplasm by active Ca²⁺-ATPase, completing the rapid change of [Ca²⁺]c and the downward transmission of signals in a short time [17,21]). The shuttle of intracellular Ca²⁺ between the cytoplasm and the major Ca²⁺ stores trigger signaling cascades. Further result proved that PKC was activated continuously during 12~48 h exposure to Cd²⁺. A prolonged activation of PKC was

also occurred at 96 h in the hepatopancreas of the $Cd^{2+}$-treated crab [17]. The DAG-PKC and $IP_3$-$Ca^{2+}$ signaling pathway was activated to initiate *S.henanense* to response $Cd^{2+}$ stress.

## 4.2. Gene expression

$Cd^{2+}$ activates above signal pathways, initiating an adaptive response of the cell and generating multiple biological effects. Expression profile analysis demonstrated that mRNA levels of detoxification-related proteins (glutathione S-transferase and metallothionein isoform), anti-oxidant enzymes (superoxide dismutase), metabolism enzyme (lactate dehydrogenase), ATP synthesis-related enzymes (cytochrome c oxidase subunit, mitochondrial ATP synthase subunit ε, cytochrome bc1 complex subunit) were up-regulated by $Cd^{2+}$ [17,30,31]. $Cd^{2+}$ usually destruct protein structure, resulting in a collection of denatured proteins, which, in turn, would stimulate the expression of chaperone proteins to repair the misfolded proteins or to eliminate the aggregated proteins via the ubiquitin-proteasome system [32]. But gene expressions of folding-related protein in gill is still unknown in *S.henanense* exposed to $Cd^{2+}$ stress. CYP450 system, the most important toxicant metabolizing enzyme families, catalyze the metabolism of various toxic substances, convert them into non-toxic metabolites [33]. Ninety percent of exogenous poisons are chiefly metabolized and detoxified via CYP450 [34]. PDI is located in the ER cavity and catalyze the formation, reduction and isomerisation of disulfide bonds between appropriate cysteine residues, which is essential for correct folding, stability of many proteins. Moreover, PDI, as a molecular chaperone, has the ability to bind to polypeptide chains, increasing the yield of correctly folded proteins [3]. Hsps are chaperones highly conserved across species that are constitutively expressed facilitating the folding, transport and assembly of polypeptides. Different stress conditions further induce their expression, maintaining protein integrity and correcting protein folding in the cell [35,36]. Hsp27 aids in refolding of nonnative proteins, preventing their non-specific aggregation and allowing them to be subsequently restored to their native structure in co-operation with ATP-dependent Hsp70 [37]. Hsp27 may be phosphorylated by PKCδ isoform after exposure to stress, followed by a subsequent increase in the expression levels of the protein within several hours [37]. BAG3, a co-chaperone, modulates ATP turnover and the protein folding activity of Hsp70 [38]. And it also interacts with HspB and acts as a scaffold that links Hsp70 with HspB, regulating degradation of ubiquitinated proteins via the proteasome and autophagy pathways [39]. Wang et al. [33] reported that $Cd^{2+}$ could affect the activity of CYP450 enzyme and gene expression in liver injury. Two-dimensional gel electrophoresis revealed that the over-expression of a PDI isoform in gills of *Eriocheir sinensis* chronically exposed to $Cd^{2+}$ could protect and/or repair target proteins and reduce the toxicity of the metal [3]. Expression of Hsp70 in hepatopancreas of crab were significantly induced by $Cd^{2+}$ [30]. Acute exposure to $Cd^{2+}$ upregulated Hsp27 and Hsp70 in transcription levels while lightly influenced the expression level of Hsp21 in the aquatic midge [36]. Hsp70 expression was up-regulated when bivalve mollusks were exposed to low concentrations of metals [35]. Expression of Hsps is associated with the concentration and the duration of $Cd^{2+}$ and cell types. In this study, the $Cd^{2+}$ treatment increased CYP450, PDI, BAG3, Hsp21, Hsp27 and Hsp70 transcription levels in gill of *S.henanense*. PDI transcript activation indicated that $Cd^{2+}$ induced ER stress and the unfolded protein response (UPR). And the increasing of PDI expression improved the synthesis and folding of proteins in rER, reduced the $Cd^{2+}$-induced cytotoxicity and was benifit to cell survival. The up-regulation of CYP450 and chaperones not only involve in protein folding but also are required to activate autophagy to degrade proteotoxic peptides, which were confirmed by the appearance of autolysosomes and the genes up-regulation involved in phagocytosis [4,30,40].

The ability of Hsp27 to aid the recovery of stress-induced denaturation of proteins may increase the survival rate of cells by limiting the levels of misfolded proteins. Hsp27-mediated activation of Akt is likely to contribute to increase the resistance to reduce apoptosis. Hsp27 negatively regulates the activation of pro-caspase-9 by interacting with cytochrome c, thus preventing the correct formation/function of the apoptosome complex [37]. Induced Hsp70 affects apoptosis at various levels through inhibiting caspase activation by interfering with Apaf-1 and preventing the recruitment of procaspase-9 to the apoptosome. Hsp70 also increases Bcl-2 expression and inhibits cytochrome C release [41]. Without doubt the expressions of CYP450, PDI, BAG3, Hsp21, Hsp27 and Hsp70 serve as a protective mechanism to increase cellular

survival during times of Cd$^{2+}$ stress. The up-regulation of these genes transcriptions avoids protein misfolding and dys-function in the stress response, while their expression decrease has been described as a possible mechanism in the cell damage or death. Cd$^{2+}$ induced Aβ and phosphorylated Tau proteins generation, mediated partially by Hsp27 and Hsp70 disruption, leading to cell death in SN56 cholinergic neurons [42]. If excessive amounts of damaged proteins are present, they would form larger aggregates which serve as the signal for the induction of apoptosis [37]. Cd$^{2+}$ led to apparent dis-tension, twisting and fracture of rER [4]. A large number of ribosomes were detached from the surface of the rER, which is is the main reason contributing to the gradual decrease in the expression level of several genes (CYP450, PDI, BAG, Hsp27, and Hsp70) after 48 h.

### 4.3. Histological damage and cell death

Cd$^{2+}$ causes pathological change in gills including thickened gill filament and enlarged gill cavity. Apical microvilli of the epithelial cells separated from the cuticle and bent, indicating that the structure of microvilli was damaged and respiratory function of gill damaged obviously. Cao et al. [43] reported that Cd$^{2+}$ disrupted tight junctions of lung epithelial cells, impair-ing epithelial barrier function, which is due to the PKC hyperphosphorylation on the connective proteins. The ultrastructure of gill further showed that the expansion of the gill cavity is closely related to the death of epidermal cells at the junction sites of the upper and lower monolayer cell [19]. After be exposeing to Cd$^{2+}$ for 48 h, nuclear of some epithelial cells in gill shrunk, became smaller and even disappeared (Fig 5B), and DNA was regularly degraded (Fig 6A), which were character-istic hallmarks of apoptosis. The apoptotic progress was accompanied by the decrease of the expression of the DPI, BAG3, Hsp27 and Hsp70. When the fresh crabs were exposed to Cd$^{2+}$ in high concentration, due to the decrease of detoxifying protein such as DPI, BAG3 and Hsps, as well as the accumulation of denatured protein, cells were unable to repair the damage caused by Cd$^{2+}$ and initiated apoptosis pathways such as mitochondrial pathways, activating caspases and causing cell apoptosis. This result has also been proven by previous studies that acute Cd$^{2+}$ exposure activated Ca$^{2+}$ signal pathway including the increases of [Ca$^{2+}$]c, CaM content and Ca$^{2+}$-ATPase activity, and triggered caspases-dependent cell apoptosis [21]. Apoptosis minizes damage to surrounding tissues through cleaning damaged cells, which is a self-protection mecha-nism and plays a crucial role in organism homeostasis and adaption to adverse stress conditions [44].

### 4.4. ATP disruption

There is an overlap between apoptosis and necrosis in Cd$^{2+}$-induced cell death in gill. However, the main internal factors that determine the mode of cell death when subjected to Cd$^{2+}$ stress are not yet understood. One evident physiological difference in cells undergoing apoptosis or necrosis is the intracellular levels of ATP [45]. Cd$^{2+}$ induces either apoptosis or necrosis in rat cortical neurons, depending on the Cd$^{2+}$ concentration [46]. Cd$^{2+}$ concentrations which activated caspase-3 and induced apoptosis in cortical neurons did not influence the intracellular ATP level; however, higher Cd$^{2+}$ concentration (10 μM) lead to both a total depletion of the intracellular ATP level and necrosis, but did not increase caspase-3 activity [46]. Mitochondria is the primary energy-generating system in most eukaryotic cells, in which 95% of the energy is synthe-sized. The integrity of inner membrane, ATP synthase with suitable activity and complete electron transport chain in inner membrane are necessary for energy synthesis. In additional, the intermembrane space of mitochondria maintain the mito-chondrial membrane potential (ΔΨm), avoiding opening of the permeability transition pore and the release of cytochrome c. Mitochondrias participate in calcium signaling and apoptosis, and their function are essential for the viability of cells. In HK-2 cell, CdCl$_2$ caused mitochondrial respiratory chain dysfunction, the decrease of both ATP content and mitochondrial membrane potential [47]. In the gills of *Eriocheir sinensis* exposed to Cd$^{2+}$, mitochondria was broken, and some enzymes involving in ATP supply were down-regulated, including GAPDH, malate dehydrogenase, ATP synthase beta and arginine kinase [3]. Mitochondrial respiration was inhibited by 30%, ATP turnover was reduced, substrate oxidation was impaired when rainbow trout was exposed to Cd$^{2+}$ [48]. Also, the necrosis induced by Cd$^{2+}$ which was accompanied by ATP disrup-tion has been found in several cell types [18,46]. In this experiment, Cd$^{2+}$ exposure for 48 h induced cell apoptosis and

decreased ATP level by 15%. After 96 h, Cd²⁺ exposure caused necrosis of gill cells and decreased ATP content by 51%. Mitochondria destruction, ATP deletion, and accumulation of misfolded proteins were the keys to cell death. Intracellular energy levels and mitochondrial function are rapidly compromised in necrosis, but not in apoptosis [49]. This may be related to the occurrence of massive cell death during the necrosis process.

## 4.5. Cd²⁺-activated signal pathway

Summing up previous study and our study, we thought Cd²⁺-stimulated PLC activation led to the decomposition of PIP$_2$ into DAG and IP$_3$, double signal moleculars. The combination of IP$_3$ in the cytoplasm and IP$_3$ receptor (IP$_3$R) on the rER by diffusion opened the Ca²⁺ channel, promoting Ca²⁺ in the ER to flow into the cytoplasm along the concentration gradient, which switch on Ca²⁺ signal pathway in cytoplasmic. In the early stages of Cd²⁺ exposure(< 48 h), interaction of Ca²⁺ with DAG activates PKC, by which active PKC phosphorylates transcription factors API or NF-κB, or activates S6 kinase or Raf kinase, promoting the gene expressions of detoxification-related proteins including BAG3, Hsps, CYP450 and DPI at transcription level, and cell survival. In the middle stages of Cd²⁺ exposure (48 h-72 h), Ca²⁺-CaM activate the caspase cascade system through the mitochondrial pathway, leading to DNA degradation and cell apoptosis. In the later stages of Cd²⁺ exposure (>72 h), due to the destruction of membrane structure and the reduction of enzyme activity, the ability of mitochondria to synthesize ATP is sharply reduced. Low ATP content and the accumulation of a large number of misfolded proteins in cells cannot maintain a series of metabolic activities, at last a large number of cells undergo necrosis (Fig 7).

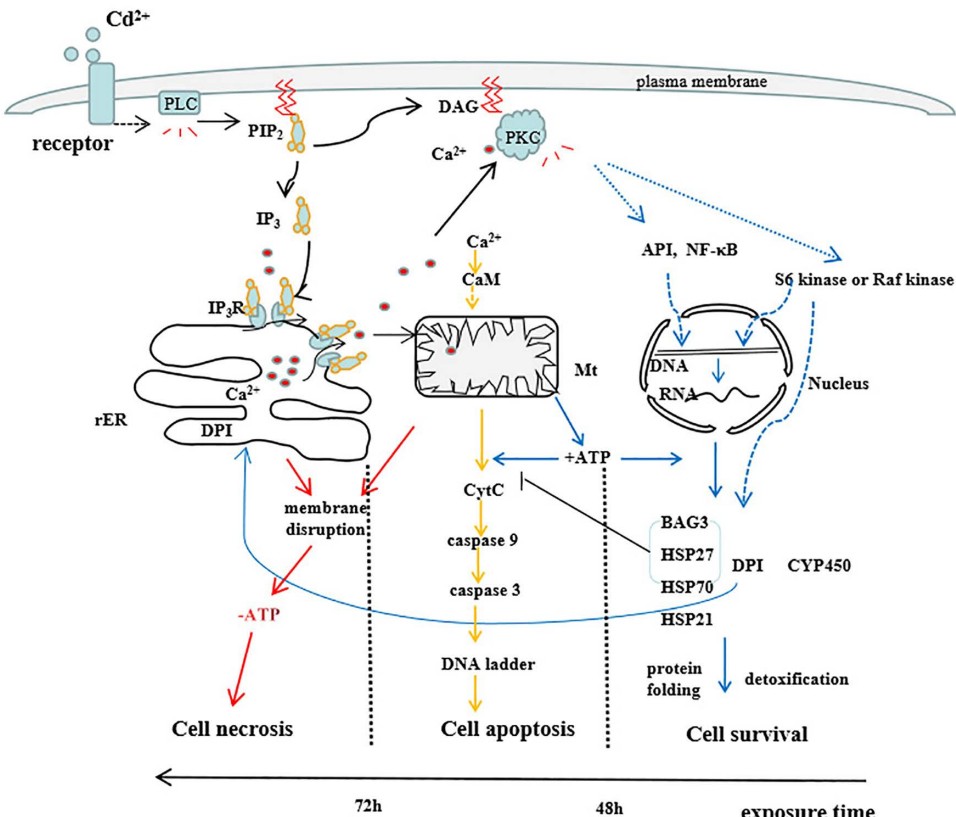

**Fig 7. Signaling cascades in gill of *S.henanense* induced by Cd²⁺.** (rER: rough endoplasmic reticulum, Mt: mitochondria, Cytc: cytochrome **c**).

## 5. Conclusions

In this study, we proved that $Cd^{2+}$ exposure firstly triggered DAG-PKC/$IP_3$-$Ca^{2+}$ double signaling pathway and promoted gene expression relating protein-folding. Moreover, based on histological observation and DNA electrophoresis, $Cd^{2+}$ exposure induced the cell injury and apoptosis in *S. henanense* gill at 48h, and caused cell necrosis accompanied by a rapid decrease in ATP at 96 h. Destruction of DNA and decrease of ATP might be the key events in the processes of cell apoptosis and necrosis. This study provided more information to know the signaling events underlying $Cd^{2+}$-induced cytotoxicity.

## Supporting information

**S1 Data. Experimental data.**
(XLS)

## Author contributions

**Conceptualization:** Jianxia Liu, Lijuan Yu, Rui Zhang.

**Funding acquisition:** Jianxia Liu.

**Investigation:** Jiachuan Yang.

**Project administration:** Jing Bai.

**Software:** Jiachuan Yang, Jing Bai.

**Supervision:** Lijuan Yu, Rui Zhang, Jing Bai.

**Validation:** Jianxia Liu, Lijuan Yu.

**Visualization:** Lijuan Yu.

**Writing – original draft:** Jianxia Liu, Lijuan Yu, Jinxiang Wang.

**Writing – review & editing:** Lijuan Yu, Jinxiang Wang.

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
