## [Decision Letter · Decision Letter 0]

31 Jul 2025

Dear Dr. Yang,

Thank you for submitting your manuscript to PLOS ONE. After careful consideration, we feel that it has merit but does not fully meet PLOS ONE’s publication criteria as it currently stands. Therefore, we invite you to submit a revised version of the manuscript that addresses the points raised during the review process.

We look forward to receiving your revised manuscript.

Kind regards,

Mei Li

Academic Editor

PLOS ONE

Journal Requirements:

“The work was supported by the Basic Research Program Project of Shanxi Province (20210302124068); Research Funding Project for Returned Overseas Students of Shanxi Province (2021-145), and Applied Basic Research Project of Datong (2022053); Shanxi Province Science Foundation for Youths [Project ID: 20240302121202].”

4. We note that your Data Availability Statement is currently as follows: All relevant data are within the manuscript and in Supporting Information files.

6. Please include your tables as part of your main manuscript and remove the individual files. Please note that supplementary tables (should remain/ be uploaded) as separate "supporting information" files.

Reviewers' comments:

Reviewer's Responses to Questions

**Comments to the Author**

1. Is the manuscript technically sound, and do the data support the conclusions?

Reviewer #1: Partly

Reviewer #2: Yes

2. Has the statistical analysis been performed appropriately and rigorously?

Reviewer #1: Yes

Reviewer #2: Yes

3. Have the authors made all data underlying the findings in their manuscript fully available?

Reviewer #1: Yes

Reviewer #2: Yes

4. Is the manuscript presented in an intelligible fashion and written in standard English?

Reviewer #1: Yes

Reviewer #2: Yes

Reviewer #1: The authors investigated Cd-induced DAG-PKC/IP3-Ca2+ double signaling pathway and cell injury in gill of freshwater crabs Sinopotamon henanense. The authors systematically designed the work by performing some valuable experimental works accordingly. It is also necessary to evaluate new data critically and not make hasty conclusions that may lead to misinterpretations. However, several important points must be addressed before publication in this journal. Also, the authors need to address all points in the revision stage for a broad range of readers' understanding.

Specific comments:

1. The introduction mentions the toxicity of cadmium and its association with organ injury, but does not specify the exposure levels of Cd in the aquatic environment. It is recommended to add "the current exposure levels of cadmium in the aquatic environment" to highlight the scientific value of this study.

2. The introduction should explain the reason for choosing the freshwater crab Sinopotamon henanense, such as "this species is a regional dominant species, sensitive to heavy metals, and is often used for toxicity assessment in aquatic environments." to enhance the rationality of the experimental design.

3. The exposure concentrations (14.5, 29, 58 mg/L) and duration (4 d) set in the Materials and Methods lack justification. It needs to be clarified whether the concentrations refer to the actual pollution levels in the environment (such as the measured concentration range of cadmium-contaminated water bodies) or preliminary experimental results.

4. In the preparation of samples on page 5, line 88 of the main text, explain why the exposure time of DAG, IP3, PKC, RNA, and histological observation for each indicator detecting Cd in the samples is different, and provide an explanation in the original text to enhance the rationality of the experimental design.

5. Is a sample size of (n=3) reasonable in the statistical analysis on page 7, line 137 of the main text? Please verify.

6. The conclusion in the main text should be cautious. The study measured the destruction of DNA and decrease of ATP as two indicators, but did not detect the genes related to apoptosis and necrosis pathways at the gene or protein level, and the evidence is insufficient.

7. Please note the language modifications, for example, the ℃ for line 117 is not Times New Roman; there are errors in the references in line 252, etc.

8. Please improve the clarity of the images and the aesthetics of the layout. The layout of Figure 2 is not very attractive; please reformat it. Figure 3 only has one image; it is recommended to combine it with other suitable images. In Figure 5A, the arrow marking for GC was not found. Additionally, the markings for Co and GL are not very clear; it is suggested to change the font color for better readability. It is recommended to explain the meaning of the triangle in the B/C/D images and the red arrows in the notes for Figure 5. In Figure 6, the B image does not indicate statistical differences, and the concentration of Cd2+ treatment is also not marked.

Reviewer #2: This study investigates the toxicological effects of cadmium (Cd²⁺) exposure in freshwater crabs (Sinopotamon henanense), with a particular focus on the underlying signaling pathways involved. Crabs were exposed to Cd²⁺ concentrations of 0, 14.5, 29, and 58 mg/L for four days. The results showed that Cd²⁺ exposure activates both the DAG–PKC and IP3–Ca²⁺ signaling pathways in gill tissue, as evidenced by elevated cytoplasmic Ca²⁺ levels and increased PKC activity. Furthermore, Cd²⁺ exposure also upregulated six stress-related genes (CYP450, BAG3, PDI, Hsp21, Hsp27, and Hsp70), suggesting involvement in detoxification and protein folding. Histological analysis revealed gill damage, including filament swelling and epidermal cell death. Additionally, DNA fragmentation and reduced ATP content were observed, potentially contributing to apoptosis and necrosis.

In conclusion, this study provides mechanistic insights into Cd²⁺-induced cellular injury in aquatic invertebrates, highlighting dual-pathway activation, oxidative stress, and disrupted energy metabolism as key contributors. The findings enhance our understanding of cadmium toxicity in freshwater ecosystems and support the study’s relevance to the scope of PLoS ONE. However, the manuscript requires substantial revisions prior to consideration for publication.

Specific comments and suggestions are as follows:

1. The final paragraph of the Introduction should be strengthened by clearly outlining the main objectives and emphasizing its scientific relevance. A more explicit statement of purpose will help readers appreciate the study’s contribution to the field of aquatic toxicology.

2. Important methodological details are missing in the Materials and Methods section. Please specify the number of biological replicates per treatment group, clarify whether n = 3 refers to biological or technical replicates, and provide details on DNA extraction and sample preservation protocols. Additionally, indicate whether normality assumptions (e.g., via Shapiro–Wilk tests) were tested prior to applying one-way ANOVA. The use of standard error (SE) rather than standard deviation (SD) to represent variability should also be justified.

3. The legend for Figure 3 incorrectly includes the description for Figure 4. Please revise to ensure that each figure legend accurately corresponds to its respective figure.

4. The primer sequences used for qPCR were not provide. These should be included in a supplementary table to ensure transparency and allow reproducibility.

5. The rationale for selecting 58 mg/L Cd²⁺ for qPCR and histological analyses is unclear. Please clarify whether this concentration was chosen based on observed toxicity, environmental relevance, or previous literature?

6. At 8 hours, both DAG and IP3 levels declined following initial elevation. This pattern warrants further discussion—could it be reflect negative feedback regulation or cellular adaptation? Similarly, the observed downregulation of several genes (PDI, BAG, Hsp27, and Hsp70) at 48 hours should be discussed to better contextualize these transcriptional responses and their implications for Cd²⁺-induced stress mechanisms.

7. In Section 3.5, the current HE-stained images lack sufficient resolution and clarity. Higher-magnification, better-focused images are recommended to improve visualization of pathological changes in gill tissues.

8. Although the Discussion provides a detailed narrative, it lacks clear structure and thematic focus. Introducing subheadings such as “Oxidative Stress,” “Gene Expression,” and “Histological Damage”would improve readability and organization. A more concise and logically structured synthesis of the proposed toxicological mechanisms would further strengthen this section.

.

Reviewer #1: No

Reviewer #2: No

---

## [Author Response · Author response to Decision Letter 1]

25 Sep 2025

Dr. Jinxiang Wang

College of Agriculture and Life Science,

Shanxi Datong University,

1 Xingyun Street, Datong, Shanxi, 037006

CHINA

Email: wangjx11@163.com

Dear Dr. editor

Thank you very much to send us the useful comments. We have carefully revised our manuscript entitled "Cadmium-induced DAG-PKC/IP3-Ca2+ double signaling pathway and cell injury in gill of freshwater crabs Sinopotamon henanense " following the comments provided by the reviewer and editor. Details on the revisions are listed below. Thank you very much again for considering our manuscript. Please do not hesitate to contact us if you have any questions. We’re looking forward to further news from you.

Reviewer # 1: Review PLOS One

Ms number: PONE-D-25-30435R1

The authors investigated Cd-induced DAG-PKC/IP3-Ca2+ double signaling pathway and cell injury in gill of freshwater crabs Sinopotamon henanense. The authors systematically designed the work by performing some valuable experimental works accordingly. It is also necessary to evaluate new data critically and not make hasty conclusions that may lead to misinterpretations. However, several important points must be addressed before publication in this journal. Also, the authors need to address all points in the revision stage for a broad range of readers' understanding.

Specific comments:

1. The introduction mentions the toxicity of cadmium and its association with organ injury, but does not specify the exposure levels of Cd in the aquatic environment. It is recommended to add "the current exposure levels of cadmium in the aquatic environment" to highlight the scientific value of this study.

Answer: Due to the thriving coal mining, coking plant, iron, steel, and chemical industries, highly toxic Cd2+ have been frequently detected in rivers, estuaries, and near-shore waters. During the heavy metal pollution incident in Longjiang, Guangxi, China in 2012, Cd2+ concentration in the water reached a peak that exceeded the standard by 80 times, and the length of the river section where Cd2+ concentration exceeded the standard by more than 5 times was 100 kilometers. We added the above descriptions to introduction (1th paragraph).

2. The introduction should explain the reason for choosing the freshwater crab Sinopotamon henanense, such as "this species is a regional dominant species, sensitive to heavy metals, and is often used for toxicity assessment in aquatic environments." to enhance the rationality of the experimental design.

Answer: Thank you for your kind suggestion. We added the above descriptions to introduction (3th paragraph).

3. The exposure concentrations (14.5, 29, 58 mg/L) and duration (4 d) set in the Materials and Methods lack justification. It needs to be clarified whether the concentrations refer to the actual pollution levels in the environment (such as the measured concentration range of cadmium-contaminated water bodies) or preliminary experimental results.

Answer: In our previous studies, we found that the 96 h LC50 value of cadmium to S. henanense is 232 mg/L for adult male crabs with a homogeneous weight (20.0±0.5g) (Wang et al., Chemosphere 2008). Considering the serious deterioration of the ecological environment, we choose relatively high Cd2+ concentrations of 14.5, 29 and 58 mg/L, consulting the 96h LC50 for Cd2+ to S. henanense. We added the above descriptions to material and methods (5th paragraph)

4.In the preparation of samples on page 5, line 88 of the main text, explain why the exposure time of DAG, IP3, PKC, RNA, and histological observation for each indicator detecting Cd in the samples is different, and provide an explanation in the original text to enhance the rationality of the experimental design.

Answer: After Cd2+ exposure, second messengers in cells are rapidly generated following activation of membrane receptors. Next, some genes are transcribed and translated through the activation of a series of signaling proteins, such as PKC phosphorylation. Ultimately, the biological effects of the cell, including changes in tissue structure and DNA degradation, are manifested. Therefore, in this experiment, DAG and IP3 were detected in the early stage of the experiment (1–8 h), PKC and RNA transcription were detected at 12–48 h, while observations of tissue structure, DNA integrity, and ATP content were detected at 48–96 h, respectively. We added the above descriptions to sample preparation.

5. Is a sample size of (n=3) reasonable in the statistical analysis on page 7, line 137 of the main text? Please verify.

Answer: In biostatistics, the requirement for sample size in analysis of variance (ANOVA) must be comprehensively determined based on the type of experimental design, data variability, effect size, and statistical power. The core objective is to ensure the results have statistical reliability and biological significance. For one-way ANOVA, the sample size per group can be relatively small, and the determination of sample size should focus on “whether the true differences between groups can be accurately detected”. In this experiment, one-way ANOVA is used to compare differences between groups, and experiments were repeated three times with similar results.

6. The conclusion in the main text should be cautious. The study measured the destruction of DNA and decrease of ATP as two indicators, but did not detect the genes related to apoptosis and necrosis pathways at the gene or protein level, and the evidence is insufficient.

Answer: We accepted the reviewer’s suggestion and have revised the discussion and conclusions. Please see “Discussion and Conclusion”.

7. Please note the language modifications, for example, the ℃ for line 117 is not Times New Roman; there are errors in the references in line 252, etc.

Answer: We accepted the reviewer’s suggestion and have revised the total manuscrip.

8. Please improve the clarity of the images and the aesthetics of the layout. The layout of Figure 2 is not very attractive; please reformat it. Figure 3 only has one image; it is recommended to combine it with other suitable images. In Figure 5A, the arrow marking for GC was not found. Additionally, the markings for Co and GL are not very clear; it is suggested to change the font color for better readability. It is recommended to explain the meaning of the triangle in the B/C/D images and the red arrows in the notes for Figure 5. In Figure 6, the B image does not indicate statistical differences, and the concentration of Cd2+ treatment is also not marked.

Answer: In accordance with the reviewers' comments, we have revised Figures 2，5 and 6, and the figure legends have also been modified accordingly. Please see “Figures” and “Figure captions”

Reviewer #2: This study investigates the toxicological effects of cadmium (Cd²⁺) exposure in freshwater crabs (Sinopotamon henanense), with a particular focus on the underlying signaling pathways involved. Crabs were exposed to Cd²⁺ concentrations of 0, 14.5, 29, and 58 mg/L for four days. The results showed that Cd²⁺ exposure activates both the DAG–PKC and IP3–Ca²⁺ signaling pathways in gill tissue, as evidenced by elevated cytoplasmic Ca²⁺ levels and increased PKC activity. Furthermore, Cd²⁺ exposure also upregulated six stress-related genes (CYP450, BAG3, PDI, Hsp21, Hsp27, and Hsp70), suggesting involvement in detoxification and protein folding. Histological analysis revealed gill damage, including filament swelling and epidermal cell death. Additionally, DNA fragmentation and reduced ATP content were observed, potentially contributing to apoptosis and necrosis.

In conclusion, this study provides mechanistic insights into Cd²⁺-induced cellular injury in aquatic invertebrates, highlighting dual-pathway activation, oxidative stress, and disrupted energy metabolism as key contributors. The findings enhance our understanding of cadmium toxicity in freshwater ecosystems and support the study’s relevance to the scope of PLoS ONE. However, the manuscript requires substantial revisions prior to consideration for publication.

Specific comments and suggestions are as follows:

1. The final paragraph of the Introduction should be strengthened by clearly outlining the main objectives and emphasizing its scientific relevance. A more explicit statement of purpose will help readers appreciate the study’s contribution to the field of aquatic toxicology.

Answer: We accepted the reviewer’s suggestion and have revised the final paragraph of the Introduction.

2. Important methodological details are missing in the Materials and Methods section. Please specify the number of biological replicates per treatment group, clarify whether n = 3 refers to biological or technical replicates, and provide details on DNA extraction and sample preservation protocols. Additionally, indicate whether normality assumptions (e.g., via Shapiro–Wilk tests) were tested prior to applying one-way ANOVA. The use of standard error (SE) rather than standard deviation (SD) to represent variability should also be justified.

Answer: The crabs were exposed to Cd2+ or water treatment for 96 h. The values are the means ± SE of three biological replicates per treatment group. Please see “2.10. Statistical analysis”. The extraction of sample DNA was performed using a DNA purification kit. Firstly, 20 mg of the sample was minced and lysed in a lysis buffer containing proteinase K, followed by centrifugation. The mixture was then eluted twice with an elution buffer, and subsequently, DNA was eluted using deionized water or a hydrophilic elution buffer. The above-described DNA extraction method has been included in the manuscript. Please see “2.8 DNA-fragmentation analysis”. In this experiment, DNA extraction, electrophoresis, and observation were performed directly after sample collection, and the entire experiment was conducted at room temperature. In fact, each experiments were repeated at least five times with similar results in our experiment. But the values are the the means ± SE of three biological replicates per treatment group. I am sorry that normality assumptions (e.g., via Shapiro–Wilk tests) were not tested prior to applying one-way ANOVA.

This experiment used crabs as the research subject, focusing primarily on investigating intracellular signal transduction and cellular stress responses following Cd²⁺ exposure, as well as exploring the toxic mechanism of Cd²⁺ on aquatic organisms. To ensure the reliability and authenticity of experimental data and avoid the impact of individual differences on the results, crabs with consistent body weight, size, sex, and activity level were selected as far as possible. Therefore, in this experiment, the Standard Error (SE) rather than the Standard Deviation (SD) was used to compare the reliability differences of the sample means of different groups.

3. The legend for Figure 3 incorrectly includes the description for Figure 4. Please revise to ensure that each figure legend accurately corresponds to its respective figure.

Answer: We accepted the reviewer’s suggestion and have made corresponding revisions. Please see “Results, 3.3 and 3.4 ”.

4. The primer sequences used for qPCR were not provide. These should be included in a supplementary table to ensure transparency and allow reproducibility.

Answer: The primer sequences used for qPCR are shown in Table 1.

5. The rationale for selecting 58 mg/L Cd²⁺ for qPCR and histological analyses is unclear. Please clarify whether this concentration was chosen based on observed toxicity, environmental relevance, or previous literature?

Answer: In our previous studies, the effects of continuous Cd²⁺ exposure on the histological structure of gills and cell death were found to be concentration-dependent. The abnormal histopathology was less severe in 14.5 mg/L Cd²⁺ with smaller gill cavity edema compared to the highest Cd²⁺ concentration group, which showed clearly abnormal histopathology. With increasing Cd²⁺ concentrations, crabs exposed to highest concentrations of Cd²⁺ (58 mg/L) contained a large number of apoptotic cells in the gill epithelium (Wang et al., PLoS ONE 2012; Wang et al., PLoS ONE 2013). Crabs exposed to 58 mg/L Cd²⁺ was used in the present study to analyze the cellular signaling pathways activated by Cd²⁺, including PKC activation, gene expression, changes in tissue structure and energy, DNA injury. We accepted the reviewer’s suggestion and have made corresponding revisions. Please see “Material and methods, 2.2 Sample preparation ”.

6. At 8 hours, both DAG and IP3 levels declined following initial elevation. This pattern warrants further discussion—could it be reflect negative feedback regulation or cellular adaptation? Similarly, the observed downregulation of several genes (PDI, BAG, Hsp27, and Hsp70) at 48 hours should be discussed to better contextualize these transcriptional responses and their implications for Cd²⁺-induced stress mechanisms.

Answer: We accepted the reviewer’s suggestion and have revised the discussion. Please see “Discussion, 4.1 and 4.2”.

7. In Section 3.5, the current HE-stained images lack sufficient resolution and clarity. Higher-magnification, better-focused images are recommended to improve visualization of pathological changes in gill tissues.

Answer: In our previous studies, hematoxylin-eosin (HE) staining and electron microscopy were used to demonstrate that Cd2+ induces gill damage and cell apoptosis (Wang et al., PLoS ONE 2012; Wang et al., PLoS ONE 2013). In the present experiment, we further verified that the DAG-PKC and IP3-Ca2+ signaling pathway are involved in gene expression and caspase-dependent cell apoptosis. Sorry, the ultrastructural images of cells were not included in this study.

8. Although the Discussion provides a detailed narrative, it lacks clear structure and thematic focus. Introducing subheadings such as “Oxidative Stress,” “Gene Expression,” and “Histological Damage”would improve readability and organization. A more concise and logically structured synthesis of the proposed toxicological mechanisms would further strengthen this section.

Answer: We accepted the reviewer’s suggestion and have revised the discussion section, and added the subheadings. Please see “Discussion”.

---

## [Decision Letter · Decision Letter 1]

27 Oct 2025

Cadmium-induced DAG-PKC/IP3-Ca2+ double signaling pathway and cell injury in gill of freshwater crabs Sinopotamon henanense

PONE-D-25-30435R1

Dear Dr. Jiaxiang Wang,

We’re pleased to inform you that your manuscript has been judged scientifically suitable for publication and will be formally accepted for publication once it meets all outstanding technical requirements.

Kind regards,

Mei Li

Academic Editor

PLOS ONE

Additional Editor Comments (optional):

Reviewers' comments:

Reviewer's Responses to Questions

**Comments to the Author**

Reviewer #1: All comments have been addressed

Reviewer #2: All comments have been addressed

2. Is the manuscript technically sound, and do the data support the conclusions?

Reviewer #1: Yes

Reviewer #2: Yes

3. Has the statistical analysis been performed appropriately and rigorously?

Reviewer #1: Yes

Reviewer #2: Yes

4. Have the authors made all data underlying the findings in their manuscript fully available?

Reviewer #1: Yes

Reviewer #2: Yes

5. Is the manuscript presented in an intelligible fashion and written in standard English?

Reviewer #1: Yes

Reviewer #2: Yes

Reviewer #1: (No Response)

Reviewer #2: The authors have adequately addressed my comments raised in a previous round of review. I think that this manuscript is now acceptable for publication.

.

Reviewer #1: No

Reviewer #2: No

---

## [Editor Report · Acceptance letter]

PONE-D-25-30435R1

PLOS One

Dear Dr. Wang,

I'm pleased to inform you that your manuscript has been deemed suitable for publication in PLOS One. Congratulations! Your manuscript is now being handed over to our production team.

Kind regards,

on behalf of

Dr Mei Li

Academic Editor

PLOS One